# Synthesis, Characterization, and Antibacterial Activity of Mg-Doped CuO Nanoparticles

**DOI:** 10.3390/molecules28010103

**Published:** 2022-12-23

**Authors:** Russul M. Adnan, Malak Mezher, Alaa M. Abdallah, Ramadan Awad, Mahmoud I. Khalil

**Affiliations:** 1Department of Chemistry, Faculty of Science, Beirut Arab University, Beirut P.O.Box 11-5020, Lebanon; 2Department of Biological Sciences, Faculty of Science, Beirut Arab University, Beirut P.O.Box 11-5020, Lebanon; 3Department of Physics, Faculty of Science, Beirut Arab University, Beirut P.O.Box 11-5020, Lebanon; 4Department of Physics, Faculty of Science, Alexandria University, Alexandria 21568, Egypt; 5Molecular Biology Unit, Department of Zoology, Faculty of Science, Alexandria University, Alexandria 21568, Egypt

**Keywords:** CuO nanoparticles, bi-metallic NPs, Mg-dopant, co-precipitation, antibacterial, EDX, photoluminescence

## Abstract

This study aims to investigate the effect of magnesium (Mg) doping on the characteristics and antibacterial properties of copper oxide (CuO) nanoparticles (NPs). The Mg-doped CuO NPs were fabricated by the co-precipitation method. NPs were characterized by X-ray Powder Diffraction (XRD), Transmission Electron Microscope (TEM), Energy Dispersive X-ray (EDX) analysis, Fourier Transform Infrared Spectroscopy (FTIR), and Photoluminescence (PL). Broth microdilution, agar-well diffusion, and time-kill assays were employed to assess the antibacterial activity of the NPs. XRD revealed the monoclinic structure of CuO NPs and the successful incorporation of Mg dopant to the Cu_1*−x*_Mg*_x_*O NPs. TEM revealed the spherical shape of the CuO NPs. Mg doping affected the morphology of NPs and decreased their agglomeration. EDX patterns confirmed the high purity of the undoped and Mg-doped CuO NPs. FTIR analysis revealed the shifts in the Cu–O bond induced by the Mg dopant. The position, width, and intensity of the PL bands were affected as a result of Mg doping, which is an indication of vacancies. Both undoped and doped CuO NPs exhibited significant antibacterial capacities. NPs inhibited the growth of Gram-positive and Gram-negative bacteria. These results highlight the potential use of Mg-doped CuO NPs as an antibacterial agent.

## 1. Introduction

Nanoparticles (NPs) revolutionized the industrial world. This revolution is due to their outstanding performance and remarkable optical, electrical, catalytic, and corrosion resistance, in addition to their antibacterial properties [1]. Copper oxide (CuO) has a vital role in multi-functional applications [2]. CuO is an important inorganic p-type semiconductor with a band gap of around 1.2–1.8 eV [3,4]. The most stable phases of copper oxide are cubic cuprous oxide (Cu_2_O) and monoclinic cupric oxide (CuO) [5]. Its applicability ranges between catalysis, photovoltaics, an electrode for lithium-ion batteries, solar energy conversion, supercapacitors, corrosion inhibition, antimicrobial, and anticancer applications [2,5,6].

CuO NPs serve as a good template for multi-functional applications. Its performance can be enhanced by implementing dopants into the CuO lattice. Since Mg dopants have enhanced the structural and antibacterial properties of CuO NPs, and they have a comparable ionic radius (72 pm) to the ionic radius of Cu^2+^ ions (73 pm) [7], the Mg-doped CuO NPs may be promising candidates for numerous applications, especially antibacterial applications [8,9,10].

Previous studies showed the antibacterial activity of doped CuO NPs. Doped NPs synthesized by co-precipitation revealed that the doping elements promote the release of Cu^2+^ from the doped CuO NPs [8,9,10]. Furthermore, the doped CuO NPs possess better antibacterial activity against Gram-positive bacteria than Gram-negative bacteria, especially against *S. aureus*. The 5% Mg-doped CuO NPs exhibited bactericidal activity at very low concentrations and their bacteriostatic rate reached 99.9% [7]. The high antibacterial activity of doped CuO NPs may be attributed to the inactivation of proteins in the cell wall of bacteria. This activity may be due to the binding of Cu^2+^ ions to the surface of the bacterial cell. Numerous previous studies showed the bactericidal action of doped CuO NPs, especially against *E. coli* and *S. aureus* [8,9,10]. This inhibitory action may be dependent on the structure of CuO NPs, as reported previously [8].

Doped NPs were shown to exhibit a better inhibitory effect on bacterial growth than that pure CuO NPs. However, other studies reported that pure NPs exhibited better inhibitory effects. The 5%, 7%, and 10% Mg-doped CuO NPs, at low concentrations, exhibited the same antibacterial activity as that of the pure CuO NPs at higher concentrations [10]. It was reported that the antibacterial activity of CuO NPs is enhanced by Mg^2+^ doping. The increased release of Cu^2+^ and Mg^2+^ ions from doped CuO with the increase of Mg^2+^ doping content may explain the inhibition of the growth of bacteria [7]. Besides, undoped CuO and Mg-doped CuO NPs showed considerable antimicrobial activity versus several bacterial pathogens, especially *S. aureus*, *P. aeruginosa*, and *E. coli* [10]. Furthermore, the doped NPs possess significant antibacterial activity against many bacterial isolates. Thus, such fabrications may provide a potential alternative to the standard methods of bacterial inhibition.

In addition, the capping of CuO NPs with Ethylenediamine tetra-acetic acid (EDTA) was shown to enhance the antibacterial activity. EDTA, which is a water-soluble polymer, acts by stabilizing the surfaces and modifying the growth (size) during NP synthesis. This enhances the antibacterial action of NPs, due to the ability of EDTA to reduce the size of NPs, which in turn increases the action of NPs against bacteria [11].

Here the impact of undoped and Mg-doped CuO NPs was explored on the inhibition of various bacterial isolates. Previous studies have shown that undoped CuO and Mg-doped CuO NPs showed considerable antimicrobial activity against several bacterial pathogens [10]. In this regard, the objective of this study is to investigate the impact of undoped and Mg-doped CuO NPs capped with EDTA on their structural, morphological, and inhibition capacities against various bacteria isolated from the Lebanese sewage sludge, including Gram-positive bacteria (*S. aureus* and *E. faecium*) and Gram-negative bacteria (*E. coli* and *S. maltophilia*).

## 2. Results

### 2.1. Characterization of NPs

#### 2.1.1. X-ray Diffraction

The XRD patterns of undoped and Mg-doped CuO NPs (Cu_1*−x*_Mg*_x_*O NPs) are shown in Figure 1. The patterns of the undoped CuO NPs show that all the peaks reflect the planes of monoclinic CuO, which are (110), (002), (111), ($\bar{1}$12), ($\bar{2}$02), (112), (020), (202), ($\bar{1}$13), (022), ($\bar{3}$11), (113), (311), and (004). Given that no other secondary phases, relating to impurities or the Cu_2_O phase are found. Moreover, when CuO is doped with magnesium, the diffraction peaks of CuO NPs are not altered. To assure this observation, the XRD patterns of all the samples are refined via the MAUD program (Figure 1). The refinements checked the possible formation of MgO as a secondary phase. However, all the samples are well fitted to the pure CuO phase, without any presence of MgO. The obtained patterns for undoped and Mg-doped CuO NPs are similar to the previously reported literature [7,12,13]. Lv et al. [7] synthesized Mg^2+^, Zn^2+,^ and Ce^4+^-doped CuO nanoparticles by the hydrothermal method. They obtained a pure phase of CuO in the XRD patterns with a doping concentration of less than 7%, indicating the total incorporation of the dopants into the lattice, without the formation of secondary phases. However, beyond 7%, at 10% doping percentage, MgO, ZnO, and CeO_2_ phases were formed in the CuO lattice. This further aligns with the present study, as all the prepared Mg-doped CuO NPs formed pure CuO phase, as the doping percentage ranged between 0.5% (*x* = 0.005) to 2% (*x* = 0.020).

#### 2.1.2. Transmission Electron Microscope

The morphology and size of the undoped and Mg-doped CuO NPs (Cu_1*−x*_Mg*_x_*O NPs) were determined using the Transmission Electron Microscope (TEM) technique. The TEM images demonstrate spherical NPs for the three selected samples, as shown in Figure 2. The Mg doping led to some noticeable changes in the NP morphology. The TEM images showed agglomeration for the undoped CuO NPs (*x* = 0.000), whereas, with the doping of Mg at concentrations of *x* = 0.005 and *x* = 0.020, the agglomeration of the doped CuO NPs is reduced, showing more uniform shapes, as depicted in Figure 2. The average particle sizes of the synthesized samples are determined from the particle size distribution which is extracted from the TEM images, using ImageJ software. This distribution is fitted by a Gaussian function, from which the average particle sizes are determined along with the standard deviation (SD), as shown in Figure 2. The obtained average particle size for undoped CuO NPs is 41.31 ± 1.76 nm. Upon Mg doping with *x* = 0.005, the particle size decreased to 27.14 ± 6.70 nm and increased slightly to 33.78 ± 8.54 nm with *x* = 0.020. These alterations in the average grain sizes with the increase in the concentration of Mg-doping may be attributed to the dissimilarity in Pauling electronegativity that affected the growth rate of Mg-doped CuO nanoparticles. The host Cu ions have a Pauling electronegativity of 1.9, which is higher than that of the doped Mg ions (1.31). This dissimilarity proves the decrease in the growth rate at low concentrations of Mg-doped CuO NPs [13]. However, at higher concentrations, the Mg-doped ions may incorporate into the lattice, not only filling substitutional sites but also occupying interstitial sites, that yield larger grains, as seen in the sample with *x* = 0.020.

#### 2.1.3. Scanning Electron Microscope and Energy Dispersive X-ray

The morphology of the nanoparticles is further studied by the scanning electron microscope, as shown in Figure 3. The SEM micrographs assure the nanocrystalline nature of the undoped and Mg-doped CuO NPs, without agglomeration. It is also noticed that the average grain size decreased with Mg-doping, re-assuring the TEM analysis. The average grain size, extracted from the SEM images, is found to be around 25.7 nm for undoped CuO NPs and 22.5 nm for Mg-doped CuO NPs. The chemical composition was studied with an energy-dispersive X-ray (EDX) technique. The presence of copper (Cu) and oxygen (O) elements in the undoped CuO NPs is confirmed by the EDX pattern, shown in Figure 3 (*x* = 0.000). No traces of precursors were detected. The EDX pattern of Mg-doped CuO NPs with *x* = 0.020 is exhibited in Figure 3. In addition to Cu and O, Mg is detected in the pattern, confirming the presence of the Mg-doped in the CuO NPs. The average atomic percent (at%) of copper and oxygen were microstructures of three different regions of the samples and demonstrated in the insets of (Figure 3) as pie charts. In undoped CuO NPs, the ratio between Cu and O is 0.82 while it is equal to 0.97 in Mg-doped CuO NPs (*x* = 0.020). This indicates that the stoichiometric nature of the samples is affected by Mg dopants. These variations may be due to surface crystalline defects [14]. Moreover, the ratio of the atomic percentage of Mg/Cu for *x* = 0.020 samples is calculated to be 0.0255, further confirming the successful synthesis of Mg-doped CuO with matching experimental and theoretical values.

#### 2.1.4. Fourier Transform Infrared (FTIR)

The FTIR spectra, represented in Figure 4, of the undoped and Mg-doped CuO NPs, demonstrated different vibrational bands. A broad absorption band ranged between 3200 and 3600 cm^−1^. The adsorbed water is present in all spectra. A small peak is detected at 2344–2354 c, and a peak centered around 1618–1656 cm^−1^ is observed. The main peaks, ranging between 700–400 cm^−1^, are displayed in Figure 4. Three peaks are detected, which are centered around 480–486, 521–530, and 580–584 cm^−1^.

It is noticed that the peak position is slightly affected by Mg concentration in the CuO lattice. The peak attributed to the symmetric vibration of Cu–O fluctuated around 483 ± 3 cm^−1^ with Mg-doping concentrations. However, Cu–O asymmetric stretching and wagging peaks of the pure CuO Nps are shifted monotonously to lower wavenumber with the Mg-doping from 530 and 584 cm^−1^ to 521 and 580 cm^−1^ with *x* = 0.020. Pramothkumar et al. [15] reported the same pattern of variation upon Mn, Co, and Ni doping to CuO NPs, and explained these shifts according to the dopant effect, which can in turn affect the surface area and defects in the samples. Singh et al. [16] reported the successful doping of Zn to CuO, which led to the increase in Cu–O bond length in the samples with the increase of Zn dopant concentration.

#### 2.1.5. Photoluminescence (PL)

Figure 5a shows the room temperature photoluminescence (PL) emission spectra for Cu_1*−x*_Mg*_x_*O NPs with an excitation wavelength of 200 nm. A prominent UV peak appeared at 310 ± 1 nm in all samples, with the highest intensity, as compared to other peaks. Furthermore, the visible emissions in the PL spectra are deconvoluted by four Voigt functions to elucidate the origin of these emissions. The position of the fitted peaks is listed in Table 1, along with the position of the UV peak. It is noticed that the increase in the concentration of the Mg doping in CuO NPs did not affect the position of the peaks, however, it affects their intensity. This is similar to the reported literature, where the doping concentration does not affect the position of the peaks in the visible part of the PL spectra [17,18,19,20]. The deconvolution of the PL spectra of Cu_1*−x*_Mg*_x_*O NPs yielded violet (391 ± 1 nm), blue (452 ± 5.5 nm), green (536 nm), and orange-red (628 ± 5 nm) emission peaks.

### 2.2. Antibacterial Activity of the Undoped and Mg-Doped CuO NPs

#### 2.2.1. MIC and MBC

The four bacterial isolates were tested for their susceptibility against the undoped and Mg-doped CuO NPs. Undoped CuO NPs had a bactericidal effect on Gram-positive bacteria while having a bacteriostatic effect on Gram-negative bacteria. The most sensitive bacterium was *E. faecium* (MIC = 0.375 mg/mL and MBC = 0.75 mg/mL), followed by *S. aureus* (MIC = 1.5 mg/mL and MBC = 3 mg/mL). *E. coli* and *S. maltophilia* were sensitive at the highest NP concentration used (3 mg/mL). Similarly, Mg-doped NPs exhibited bactericidal activity on Gram-positive bacteria and bacteriostatic activity on Gram-negative bacteria. *E. faecium* was the most sensitive bacterium (MIC = 1.5 mg/mL and MBC = 3 mg/mL) and *E. coli* was the most resistant bacterium. Collectively, undoped and Mg-doped NPs had a better effect on Gram-positive bacteria than that on Gram-negative bacteria. The MIC and MBC results are shown in Table 2 and Figure A1, Figure A2 and Figure A3 (Appendix A). 

It was reported that when the MBC/MIC ratio ˂ 4, this will reflect a bactericidal effect [21]. The antibacterial results against the four bacterial isolates shown in Table 2 revealed that the undoped and Mg-doped NPs exhibited bactericidal effects against *S. aureus* and *E. faecium* with MBC/MIC ratio = 2 and a bacteriostatic effect against *E. coli* and *S. maltophilia*.

#### 2.2.2. Agar Well Diffusion

All isolates showed sensitivity to the undoped CuO NP (*x* = 0.000). *E. faecium* was the most sensitive. It showed a sensitivity against the lowest NP concentration (0.1875 mg/mL). The other three bacteria showed a sensitivity against an NP concentration of 1.5 mg/mL. Sensitivity was considered positive for ZOI diameters > 7 mm [17]. All the bacterial isolates were sensitive to the Mg-doped CuO NP with *x* = 0.005. *E. coli* and *S. maltophilia* showed a sensitivity to the NP at a concentration of 0.75 mg/mL. *E. faecium* and *S. aureus* showed sensitivity against a concentration of 1.5 mg/mL. All investigated bacteria showed sensitivity to Mg-doped NPs with *x* = 0.010. *S. aureus* showed susceptibility against a concentration of 0.75 mg/mL. *E. faecium* showed a sensitivity against a concentration of 1.5 mg/mL. *S. aureus*, *E. coli*, and *S. maltophilia* exhibited a sensitivity at the highest concentration (3 mg/mL). All isolates were sensitive to the Mg-doped CuO NP with *x* = 0.015. *E. faecium* was the most sensitive. It showed a sensitivity starting from the lowest concentration (0.1875 mg/mL). *S. aureus*, *E. coli*, and *S. maltophilia* were sensitive at concentrations starting from 0.375 mg/mL. All isolates were sensitive at the highest NP concentration (3 mg/mL). For Mg-doped CuO NP with *x* = 0.020, all bacteria exhibited sensitivity only at the highest NP concentration (3 mg/mL). The agar well diffusion results of the undoped and Mg-doped CuO NPs are shown in Table 3 and Figure A4 (Appendix A). All results were significant with a *p*-value ˂ 0.05 shown in Table A1 (Appendix A).

#### 2.2.3. Time-Kill Results

The time-kill test was performed using the MICs of the undoped and Mg-doped CuO NPs against four bacterial isolates to detect the time needed for each NP to exert its antibacterial effect. All bacterial isolates were sensitive to all tested NPs after 2 h of incubation. The activities were sustained till 24 h of incubation. The time-kill results for the different bacterial isolates are shown in Figure 6.

## 3. Discussion

The peaks of the undoped CuO NPs shown by the XRD correspond to the primary defining peaks of the monoclinic CuO phase with a space group of C2/c [22]. The absence of a secondary phase suggests that the samples exhibit a highly single phase [23]. The refinements indicate the total incorporation of Mg dopant in the Cu_1*−x*_Mg*_x_*O NPs.

The TEM results demonstrated the change in the morphology and size of the NPs after doping, indicating that the increase in the doping concentration increases the uniformity of the NPs and in turn decreases their size.

The composition of the NPs was detected by EDX. The absence of precursors indicates the purity of the formed CuO NPs. Noting that the emergence of carbon may be due to the use of carbon tape in the measurements, or some residues from EDTA [24]. In addition, the ratio between Cu and O in the undoped and Mg-doped NPs indicates that the stoichiometric nature of the samples is affected by Mg dopants. These variations may be due to surface crystalline defects [14].

Infrared spectroscopy can be used as a fingerprint to identify different molecules by comparing vibration bands. The broad absorption band observed by the FTIR is associated with the hydroxyl (O–H) stretching vibration mode of the water molecule [25]. The adsorbed water observed is due to the physical adsorption of water from the atmosphere [26]. The small peak observed at 2344–2354 cm^−1^ is related to the vibration of CO_2_ in the air [26] and the peak centered around 1618–1656 cm^−1^ is due to H–O–H bending vibrations of water molecules [25]. The three main peaks are attributed to Cu–O symmetric stretching, Cu–O asymmetric stretching, and Cu–O wagging, respectively. These peaks validate the successful formation of CuO NPs [5]. Pramothkumar et al. [15] reported the same pattern of variation observed upon Mn, Co, and Ni doping to CuO NPs, and explained these shifts according to the dopant effect, which can in turn affect the surface area and defects in the samples. Singh et al. [16] reported the successful doping of Zn to CuO, which led to the increase in Cu–O bond length in the samples with the increase of Zn dopant concentration.

The PL spectra detect the imperfections and defects within the samples, where the prevalence of the imperfections and surface states varies depending on the synthesizing circumstances, particle size and shape, types of dopants, and concentrations [27,28]. The origin of the UV peak is directly related to the recombination of electron–hole pair, near the band gap transition [17,20]. It is noticed that the position of the UV peak is slightly invariant with the doping concentration, however, its intensity increased with Mg-doping. This enhancement of the intensity may be related to the passivation of surface defects that generate radiative recombination [29]. Additionally, the intensity of the UV peak is affected by the electron density and the variation of the morphology and size of the nanoparticles, with the increase of the doping concentration [30]. The visible emissions are highly sensitive to the change in the synthesis conditions, accounting for the type of dopant and its concentration, the size of the nanoparticle, and its morphology [18]. The size of Mg-doped CuO NPs decreased with the increase of the doping concentration, as noted from TEM and SEM analysis. Hence, the large surface-to-volume ratio stimulates more surface-defect states, as vacancies and interstitials, creating trap levels that radiate visible emissions [18]. Mainly, the intensity of the visible emissions is quenched with the increment of the doping concentration in Cu_1*−x*_Mg*_x_*O NPs, as can be noticed from the inset of Figure 5a. This decrement in the intensity may be due to the trapping of the photoexcited electron from the conduction band of CuO NPs by the formed deep-level centers from Mg doping [17,30]. The violet and blue emissions are mostly attributed to deep-level defects, indicating the existence of Cu vacancies in the lattice [17]. The green emission was reported to originate from the recombination of single ionized electrons with a photogenerated hole in the valence band, noting the presence of singly ionized oxygen vacancies or dangling bonds of copper [19]. The orange–red emission ascends from the recombination of an electron bound to donor and free holes [18].

The reported antibacterial properties of NPs, especially CuO NPs, made their usage efficient against bacteria [1,2]. In this regard, CuO NPs were used against four bacterial pathogens isolated from sewage sludge. All bacterial isolates, except *S. maltophilia*, are frequently present in Lebanon, especially in the feces of animals. *S. maltophilia* is a rare Gram-negative bacterium in Lebanon [21,31,32]. In this investigation, the antibiotic Dox was used as a reference antibiotic. It belongs to the tetracycline family of antibiotics. It acts by inhibiting protein synthesis by binding to the 30S ribosomal subunit, leading to the destruction of the bacterial cells [33,34,35]. This study showed that the undoped and Mg-doped CuO NPs exerted antibacterial activities against all bacterial isolates. Using the agar well diffusion assay, the NPs had better effects on Gram-positive bacteria than on Gram-negative bacteria. The results are consistent with previous studies that showed that Gram-negative bacteria are more resistant to NPs, due to the rigidity of their cell wall [36,37,38]. This activity depends on the metal oxides present in the NPs. The latter could penetrate the cell wall of bacteria, leading to cell autolysis [10]. Among the investigated NPs, the undoped and Mg-doped CuO NPs with *x* = 0.005 and *x* = 0.010 were efficient as antibacterial agents. They were effective at low NPs concentrations against all bacterial isolates. In contrast, the Mg-doped CuO NPs with *x* = 0.015 and *x* = 0.020 had lower antibacterial activity against the investigated bacterial isolates. They were effective at higher concentrations. Regarding bacterial susceptibility, *S. aureus* and *E. faecium* were the most sensitive. Their growth was inhibited by all tested NPs at significantly low concentrations. These results are consistent with previous studies that have shown the sensitivity of Gram-positive bacteria, especially *S. aureus*, to NPs [8,9,39]. On the other hand, *E. coli* and *S. maltophilia* were more resistant. The inhibition of their growth required higher concentrations of the NPs. This could be attributed to the shape of NPs. Their spherical shape, demonstrated by the TEM results reflects their significant inhibitory activity against Gram-positive bacteria [8,40]. Spherical NPs are shown to have good antibacterial activity due to the sphere prisms that can penetrate easily into the bacterial cell membrane [41].

The MIC results confirmed the results of the agar well diffusion assay. All the investigated NPs, except the Mg-doped CuO NPs with *x* = 0.020, had bactericidal activities. The Mg-doped CuO NP with *x* = 0.020 had bacteriostatic activity only. The bacteriostatic and bactericidal effects of NPs depend on their metal oxide content and their morphology, and the architecture of the bacteria [42,43]. This means that the metal oxides of the tested NPs can react with the bacterial cell wall through special mechanisms, which are still not very specific. However, previous studies reported the following mechanisms: disruption of the bacterial cell wall, generating reactive oxygen species, and binding with specific cytoplasmic targets and production of metabolites, leading to these bacteriostatic and bactericidal effects [42,44]. Kumer et al. showed that ZnO NPs exhibited good antibacterial activity against Gram-positive bacteria. This activity was better than that of Ag-doped ZnO NPs [45]. In addition, Prakash et al. reported that TiO_2_ NPs with doped antibacterial activity were better than the undoped TiO_2_ NPs [46]. The bactericidal properties of the NPs depend on the shape, the surface area of the particle, the type of metal ions, and the chemically reactive functional groups [8,10,35]. The high bactericidal effect is attributed to the different shapes (spherical in our case) of the particles, which help them penetrate the bacterial cell membrane. Moreover, the large surface area permits the production of reactive oxygen. This induces oxidative stress on bacteria, which interrupts the electron flow in the inner membrane, thus causing cell damage [35,36]. On the other hand, the observed bacteriostatic effect could be due to the low number of metal ions coming from the metal oxides, which prevents the Cu ions from interacting with the bacterial cell wall. So, the main bactericidal mechanism may rely on the damage of the cell membrane by the metal oxides [35,37].

Time-kill results have shown that the most frequent inhibition time of the tested NPs started at 2 h of incubation. NPs can prevent the adaptation and duplication of bacteria [47]. This effect could be attributed to the limiting effect of NPs on the nutrient uptake by the bacteria, which eventually will lead to cell lysis. This is consistent with previous studies that showed that NPs affect the metabolic activities and division of bacterial cells. Metal oxides may lead to nutrient deprivation [8,10,36]. In addition, the size of the particles and the surface area may specify the time needed for the interaction between the NP and the bacterial cell wall. When the size of the particle is smaller, the interaction becomes faster, thus decreasing the time needed for the inhibition of bacterial growth [8,10,36]. This slows metabolic processes and leads to cell death.

Collectively, previous studies reported that the size of the NPs reflects their antibacterial effect [48]. In addition, the variation in the antibacterial activity depends on the morphology of the NPs. Furthermore, the variation in the intensity of PL accompanied by the variation in oxygen interferes with the antibacterial activity. So, the shape and the morphology of NPs play a vital role in the inhibition of bacterial growth.

## 4. Materials and Methods

### 4.1. Synthesis of NPs

The undoped and Mg-doped CuO NPs were prepared by the co-precipitation method, with the chemical formula of Cu_1*−x*_Mg*_x_*O (*x* = 0.000, 0.005, 0.010, 0.015, and 0.020). The CuO NPs were synthesized using copper (II) chloride dehydrate (Merk), magnesium chloride hexahydrate (Sigma-Aldrich, ≥99.0%), and ethylenediamine tetra-acetic acid (EDTA) (0.1 M). The weighed reagents were prepared with a molarity of 1 M in de-ionized water and stirred for 15 minutes. The solution was then titrated with sodium hydroxide NaOH (2 M). NaOH was added slowly under vigorous stirring until pH reached 12. After that, the precipitate was heated at 60 °C for 2 h, then sonicated for 10 minutes. The black precipitate obtained was washed with de-ionized water several times until pH reached 7. Finally, the washed precipitate was dried at 100 °C for 16 h and ground into fine powders. The powders were sintered at 600 °C for 4 h.

### 4.2. Characterization of Mg-Doped CuO NPs

The structural properties of both undoped and doped CuO NPs were studied by XRD using the X-ray Bruker D8 Focus power diffractometer with Cu K_α_ radiation, operated at 40 kV and 40 mA, in the range 20 ≤ 2θ° ≤ 80. Material Analysis Using Diffraction (MAUD) software was then used to check for the presence of CuO and MgO phases in the resultant NPs using the CIF files downloaded from the Crystallography Open Database (COD). The morphology of the prepared CuO NPs was investigated using the JEM 100 CX Transmission Electron microscope (TEM), operated at 80 kV. The main functional groups of the synthesized samples were detected using the Nicolet iS5 Fourier Transform Infra-Red (FTIR) spectra after preparing potassium bromide (KBr) pellets mixed with the undoped and doped CuO NPs (1:100). The purity of the Cu_1*−x*_Mg*_x_*O NPs was studied using energy dispersive X-ray (EDX), operated at a voltage of 20 kV with laser power of 5 mW and magnification objective of 50x. The Photoluminescence (PL) spectra were studied by a Jasco FP-8600 spectrofluorometer with Xenon (Xe) laser at 200 nm excitation wavelength for Cu_1*−x*_Mg*_x_*O nanoparticles, dispersed in ethanol.

### 4.3. Isolation of Bacteria

Briefly, *E. faecium, S. aureus, E. coli, and S. maltophilia* were isolated from wastewater by streaking 100 µL of the samples on different selective media (blood agar, chocolate agar, MacConkey agar, mannitol salt agar (MSA), eosin methylene blue (EMB) agar, and cetrimide agar). The plates were incubated at 37 °C for 24 h. After isolation, bacteria were Gram stained to differentiate between Gram-positive and Gram-negative bacteria. Bacteria were further identified by VITEK assay.

### 4.4. Broth Microdilution Assay: Minimum Inhibitory Concentration (MIC) and Minimum Bactericidal Concentration (MBC)

The MICs of the undoped and Mg-doped CuO NPs were determined against four bacteria employing the microwell dilution method. The test was performed in sterile 96-well microplates by dispensing into each well 90 µL of nutrient broth and 10 µL of bacterial suspensions adjusted to 0.5 McFarland. Then, 100 µL of each NP (0.1875–3 mg/mL) was added to the wells. The plates were incubated at 37 °C for 24 h and the optical density (O.D.) was measured at 595 nm, using an ELISA microtiter plate reader. The MIC is defined as the lowest concentration of the NPs that inhibits the visible growth of the tested bacteria in the wells. Doxycycline (Dox) was used as a reference antibiotic. After incubation, 10 µL of the clear wells was transferred to Muller Hinton agar (MHA) plates and incubated at 37 °C for 24 h to detect the MBC [16]. The MBC is defined as the lowest concentration that inhibits the visible growth of bacteria on the plates. All experiments were repeated at least three times.

### 4.5. Agar Well Diffusion Assay

Agar well diffusion assays were performed in triplicate for the undoped and Mg-doped CuO NPs on four bacterial isolates using MHA. A standard inoculum was prepared for each tested bacterial isolate as described in the MIC and MBC broth microdilution assay. The plates were inoculated with 100 µL of each bacterial suspension, which was spread evenly over the entire surface of the agar. Plates were then punched with a 6 mm cork-borer. A total of 100 μL of each NP (0.1875–3 mg/mL) was pipetted into the wells and the plates were incubated at 37 °C for 24 h. Dox and Amoxicillin (Amo) were used as reference antibiotics. For each well, the diameter of the zone of inhibition (ZOI) was measured. ZOI of diameter *>* 7 mm was considered a significant inhibitory effect [49].

### 4.6. Time-Kill Test

Time-kill studies were performed to detect the time needed by the undoped and Mg-doped CuO NPs to inhibit bacterial growth. The test was performed in sterile 96-well microplates by dispensing into each well 90 μL of nutrient broth and 10 μL of the bacterial suspensions adjusted to 0.5 McFarland. Then, 100 μL of each NP’s MIC was added to the wells. The plates were incubated at 37 °C and the O.D. was measured at 595 nm, using an ELISA microtiter plate reader at different time points (0–24 h) [47]. All experiments were repeated at least three times.

### 4.7. Statistical Analysis

All statistical tests were done in Excel software, and graphs were drawn on Origin software. Statistical significance was determined by *t*-test. Differences with *p*-value ˂ 0.05 were considered statistically significant.

## 5. Conclusions

Pure and Mg-doped CuO NPs were fabricated via the co-precipitation method. The XRD patterns with their refinements confirmed the total incorporation of Mg dopant in the Cu_1*−x*_Mg*_x_*O NPs and the production of CuO NPs without any impurities. Besides, the morphology was changed upon Mg doping, in which the NPs showed a uniform shape with less agglomeration. FTIR spectra demonstrated the main vibrational modes of undoped and doped CuO NPs. The Cu–O bond was shifted as the Mg concentration for doping increased, confirming the incorporation of the dopant and its effect in modifying the surface area and defects. The EDX patterns further confirmed the purity of CuO NPs and the inclusion of Mg inside the NPs successfully. PL studies proved the enhancement of visible emissions of CuO nanoparticles associated with Mg doping. This study showed significant antibacterial activity of undoped and Mg-doped NPs. The results showed that the NPs had significant antibacterial activity against different Gram-positive and Gram-negative bacteria. Thus, the undoped and Mg-doped CuO NPs exhibited a significant impact on the structural, morphological, and inhibition capacities against *S. aureus, E. faecium, E. coli,* and *S. maltophilia,* isolated from the Lebanese wastewater. These results may provide an approach to using CuO NPs as antibacterial agents to prevent bacterial contaminations.

## Figures and Tables

**Figure 1 molecules-28-00103-f001:**
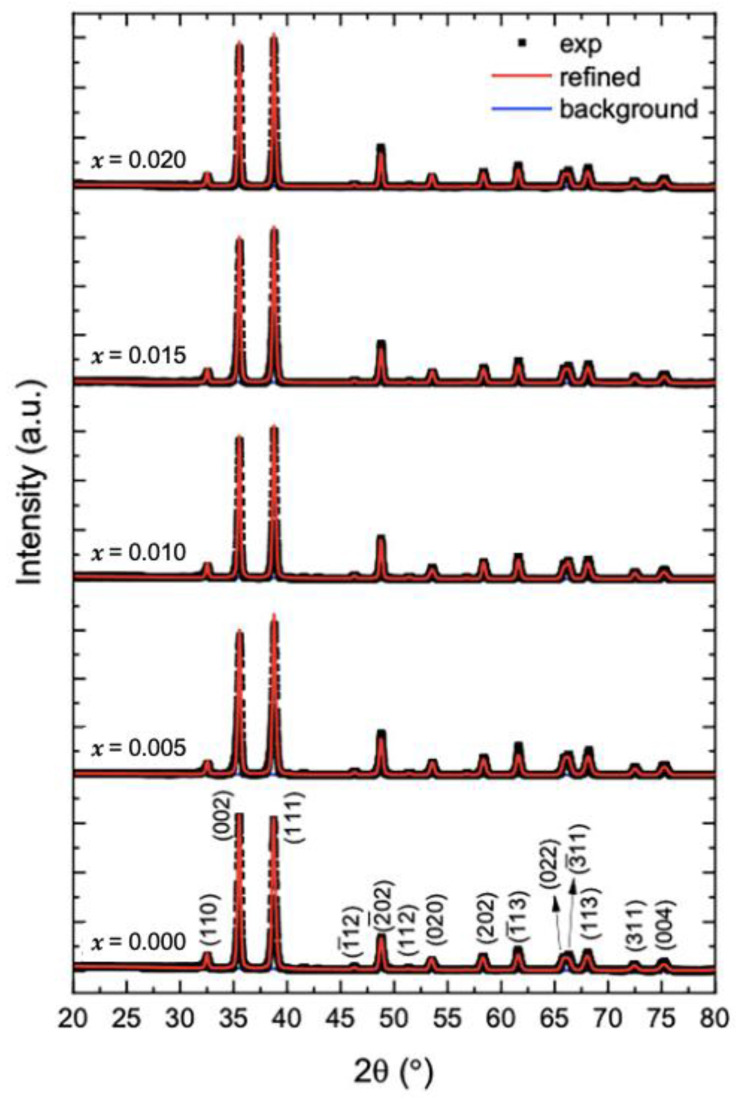
Refinements of the XRD patterns of undoped and Mg-doped CuO NPs (Cu_1*−x*_Mg*_x_*O NPs).

**Figure 2 molecules-28-00103-f002:**
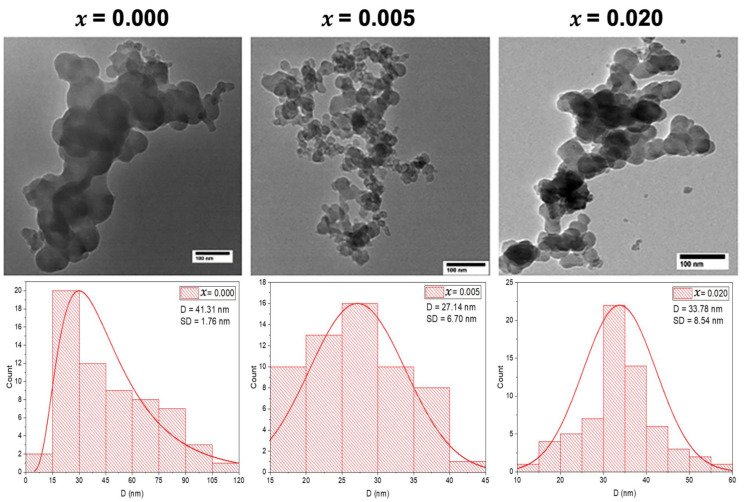
TEM images with the grain size distribution of the undoped and Mg-doped CuO NPs (Cu_1*−x*_Mg*_x_*O NPs).

**Figure 3 molecules-28-00103-f003:**
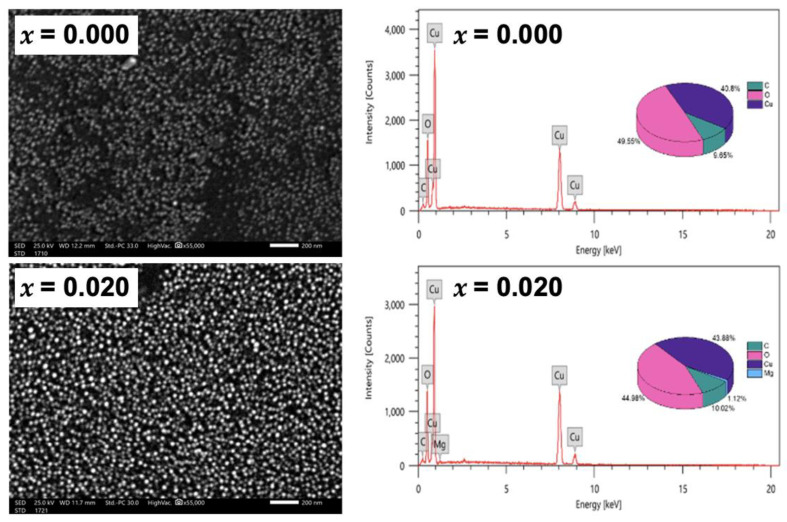
SEM and EDX patterns for undoped, and Mg-doped CuO NPs (Cu_1*−x*_Mg*_x_*O NPs).

**Figure 4 molecules-28-00103-f004:**
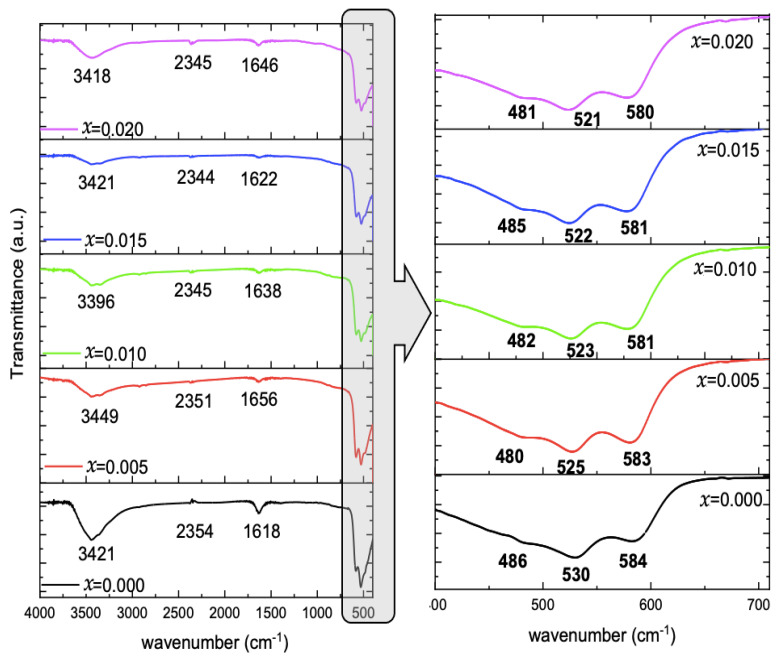
FTIR spectra of Cu_1*−x*_Mg*_x_*O NPs, and enlarged view for Cu-O vibrations.

**Figure 5 molecules-28-00103-f005:**
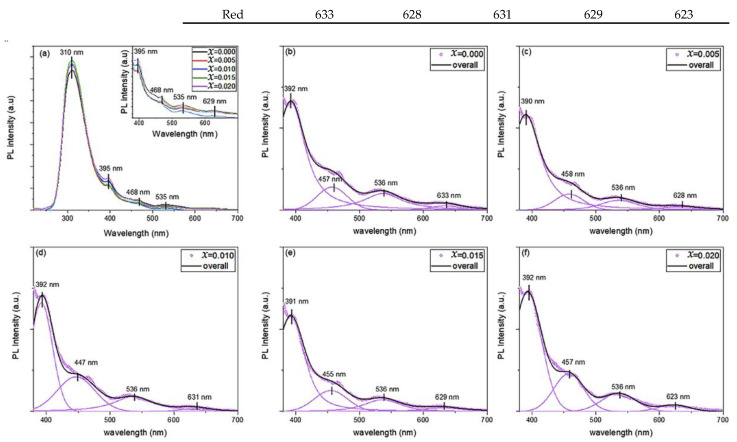
(**a**) Photoluminescence spectra of Cu_1*−x*_Mg*_x_*O NPs, and (**b**–**f**) their deconvolution for undoped and Mg-doped CuO NPs.

**Figure 6 molecules-28-00103-f006:**
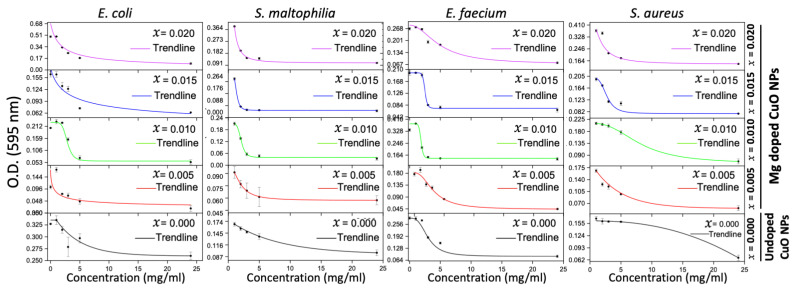
Time-kill results of the undoped and Mg-doped NPs against four bacterial isolates.

**Table 1 molecules-28-00103-t001:** Position of peaks by deconvolution of PL spectra for Cu_1*−x*_Mg*_x_*O NPs.

Peaks Color	Wavelength (nm)				
	*x* = 0.000	*x* = 0.005	*x* = 0.010	*x* = 0.015	*x* = 0.020
UV	310	310	311	309	309
Violet	392	390	392	391	392
Blue	457	458	447	455	457
Green	536	536	536	536	536
Red	633	628	631	629	623

**Table 2 molecules-28-00103-t002:** MIC and MBC of the undoped and Mg-doped (Cu_1*−x*_Mg*_x_*O) NPs against four bacterial isolates.

Bacterial Isolates	MICs and MBCs (mg/mL)														
	*x* = 0.000			*x* = 0.005			*x* = 0.010			*x* = 0.015			*x* = 0.020		
	MIC	MBC	$\frac{MIC}{MBC}$	MIC	MBC	$\frac{MIC}{MBC}$	MIC	MBC	$\frac{MIC}{MBC}$	MIC	MBC	$\frac{MIC}{MBC}$	MIC	MBC	$\frac{MIC}{MBC}$
Gram-positive bacteria															
*S. aureus*	1.5	3	2	1.5	3	2	1.5	3	2	3	˃3	nd	1.5	3	2
*E. faecium*	0.375	0.75	2	1.5	3	2	1.5	3	2	1.5	3	2	1.5	3	2
Gram-negative bacteria															
*E. coli*	3	˃3	nd	3	˃3	nd	3	˃3	nd	3	˃3	nd	3	˃3	nd
*S. maltophilia*	3	˃3	nd	1.5	3	2	1.5	3	2	1.5	3	2	3	˃3	nd

nd: not determined, MIC: minimum inhibitory concentration, MBC: minimum bactericidal concentration.

**Table 3 molecules-28-00103-t003:** Agar well diffusion of undoped and Mg-doped CuO NPs against four bacterial isolates.

Nanoparticles		Bacterial Isolates			
		Gram-Positive Bacteria		Gram-Negative Bacteria	
		*S. aureus*	*E. faecium*	*E. coli*	*S. maltophilia*
Sample	Concentration (mg/mL)	ZOI ± SEM (mm)			
*x* = 0.000	0.1875	7 ± 0.0	7.6 ± 0.2	0 ± 0.0	0 ± 0.0
	0.375	7 ± 0.0	8.6 ± 0.2	0 ± 0.0	0 ± 0.0
	0.75	7 ± 0.0	9 ± 0.7	0 ± 0.0	0 ± 0.0
	1.5	8 ± 0.4	12.3 ± 1.4	19 ± 0.9	14 ± 0.4
	3	12.3 ± 0.2	13.3 ± 0.7	22.3 ± 0.3	17.3 ± 0.4
*x* = 0.005	0.1875	7 ± 0.0	7 ± 0.0	0 ± 0.0	0 ± 0.0
	0.375	7 ± 0.0	7 ± 0.0	0 ± 0.0	0 ± 0.0
	0.75	7 ± 0.0	7 ± 0.0	10 ± 0.0	8.3 ± 0.2
	1.5	9.6 ± 0.2	8 ± 0.4	15.6 ± 0.7	10.6 ± 0.2
	3	13.6 ± 0.2	12.3 ± 0.2	20 ± 0.4	11.6 ± 0.5
*x* = 0.010	0.1875	7 ± 0.0	7 ± 0.0	0 ± 0.0	7 ± 0.0
	0.375	7 ± 0.0	7 ± 0.0	0 ± 0.0	7 ± 0.0
	0.75	7.6 ± 0.2	7 ± 0.0	0 ± 0.0	7 ± 0.0
	1.5	7.6 ± 0.2	8.6 ± 0.5	0 ± 0.0	7 ± 0.0
	3	10 ± 0.9	10.6 ± 0.7	16.6 ± 0.7	7 ± 0.0
*x* = 0.015	0.1875	7 ± 0.0	7.6 ± 0.2	0 ± 0.0	0 ± 0.0
	0.375	7 ± 0.0	7.6 ± 0.2	0 ± 0.0	0 ± 0.0
	0.75	10.3 ± 0.2	7.6 ± 0.2	0 ± 0.0	0 ± 0.0
	1.5	11.3 ± 0.2	7.6 ± 0.2	11.6 ± 0.5	0 ± 0.0
	3	17.6 ± 1.18	7.6 ± 0.2	16.3 ± 0.9	10.6 ± 0.5
*x* = 0.020	0.1875	0 ± 0.0	0 ± 0.0	0 ± 0.0	0 ± 0.0
	0.375	0 ± 0.0	0 ± 0.0	0 ± 0.0	0 ± 0.0
	0.75	0 ± 0.0	0 ± 0.0	0 ± 0.0	0 ± 0.0
	1.5	0 ± 0.0	0 ± 0.0	0 ± 0.0	0 ± 0.0
	3	9.6 ± 0.7	9.3 ± 0.5	13.6 ± 0.7	7.6 ± 0.2
Dox	0.25	33.6 ± 0.7	19.3 ± 0.7	27.6 ± 0.2	34.6 ± 0.2
Amo	0.25	0 ± 0.0	0 ± 0.0	0 ± 0.0	0 ± 0.0

Dox: Doxycycline, Amo: Amoxicillin, ZOI: zone of inhibition, SEM: standard error of the mean.

## Data Availability

The data supporting the reported results are available with the corresponding author and will be provided upon request.

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
