# Peer review of "Synthesis, Characterization, and Antibacterial Activity of Mg-Doped CuO Nanoparticles"

_molecules, 2022, doi:10.3390/molecules28010103_

Round 1

Reviewer 1 Report

Khalil et al. prepared the CuO NPs doped by Mg by the co-precipitation method in the presence of ethylenediamine tetra acetic acid as a stabilizing agent for use as antibacterial agent. Fabrication and research of CuO NPs for  different applications is well explored area of NP science. Considering a number of papers relating to the synthesis of CuO NPs, as well different metal doped CuO NPs,  and subsequent applications as antibacterial agents, the novelty of this manuscript is insufficient for its publication in Molecules. Moreover, in the introduction authors refer to the results published by others demonstrating antibacterial activity of CuO NPs and doped CuO NPs against a wide range of bacterial pathogens. Additionally, it was shown that activities of pure and doped CuO NPs are similar. The question arises, what the special purpose of the current manuscript?  

In general, the design of investigation is very similar to one published by  Pramothkumar, (ref. 27 of the current manuscript) and some others.

In addition to the general conclusion about the hopelessness of publication in Molecules, I would like to draw attention to the shortcomings of the study itself in terms of the methods used and the interpretation of the results.

For example, the authors do not take into account the possible content of carbon and nitrogen residues from ethylenediamine after the sintering process. Instead, a presence of small amount of carbon in EDX spectra is attributed to carbon tape, used in the measurements. I would suggest to perform TGA analysis before a sintering to elucidate the ethylenediamine decomposition process in course of calcination of nanocomposites. 

On TEM images, the size of the NPs looks very confusing. The images attributed by the authors to NPs look more like an organic layer than metal NPs. In addition, the dimensions of NP 41.31; 27.14 etc. cannot be indicated in this way. The method of calculation must be specified, as well as the margin of error. The explanation of the change in the size of nanoparticles by doping with Mg looks very strange and is not confirmed by either other methods or references.

SEM images  (Fig.3) are shown at very different scales that does not allow assessment  the similarity or difference between doped and undoped CuO NPs.

The PL spectra ( Fig.5) also look equally. The deconvolutions have been done in an arbitrary manner that have led to unreliable conclusions.

In the list of references the titles of journals are missed (see refs.  5,8,9,10.30,43,45, etc). The ref. 11 duplicates the ref. 9. In general, the references list is very sloppy  , with some data missed.

Author Response

[Molecules] Manuscript ID: molecules-2072231 – Authors Response

The authors would like to thank the reviewer for his/her precious time, efforts, and invaluable comments. The comments and suggestions were very helpful and had allowed us to explain and improve more aspects of the manuscript. We have carefully addressed all the comments and we are pleased to learn that the manuscript has been substantially improved.

Below is our point-by-point response to the comments as the following:

Reviewer Comments & Authors Responses

  • Khalil et al. prepared the CuO NPs doped by Mg by the co-precipitation method in the presence of ethylenediamine tetra acetic acid as a stabilizing agent for use as antibacterial agent. Fabrication and research of CuO NPs for different applications is well explored area of NP science. Considering a number of papers relating to the synthesis of CuO NPs, as well different metal doped CuO NPs, and subsequent applications as antibacterial agents, the novelty of this manuscript is insufficient for its publication in Moreover, in the introduction authors refer to the results published by others demonstrating antibacterial activity of CuO NPs and doped CuO NPs against a wide range of bacterial pathogens. Additionally, it was shown that activities of pure and doped CuO NPs are similar. The question arises, what the special purpose of the current manuscript?

Reply: We would like to thank the reviewer for this comment. While we understand the reviewer’s concern and agree that a wide range of bacteria was shown to be sensitive to the undoped and doped CuO NPs in previous literature. We would like to highlight the following:

  1. The current study is the first to report the effect of undoped and doped CuO NPs on the multidrug-resistant gram-negative maltophilia. To the best of our knowledge, and after searching the PubMed database using the following query: ("stenotrophomonas maltophilia"[MeSH Terms] OR ("stenotrophomonas"[All Fields] AND "maltophilia"[All Fields]) OR "stenotrophomonas maltophilia"[All Fields]) AND "maltophilia"[All Fields] AND ("s"[All Fields] AND "maltophilia"[All Fields]) AND "CuO"[All Fields] AND ("nanoparticle s"[All Fields] OR "nanoparticles"[MeSH Terms] OR "nanoparticles"[All Fields] OR "nanoparticle"[All Fields]), Only one very recent study showed up which did not investigate the effect of CuO NPs on S. maltophilia, but investigated the capability of S. maltophilia, isolated from the soil, to synthesis CuO and ZnO nanoparticles [1].
  2. The same bacterial species may differ between countries and even between two different areas in the same country. The bacteria used in the current study were isolated from wastewater as mentioned in the Material & Methods section. And to the best of our knowledge, no studies have been carried out to study the effect of NPs on bacteria isolated from wastewater.
  3. According to an elegant recent study by Van Rossum et al. (2020) [2], Many species have been found to have both pathogenic and commensal strains (for example, Escherichia coli and Bacteroides fragilis). Indeed, a classic example are coli strains, which can be pathogenic, commensal, host-associated or environmental. The hierarchy of terms used with bacteria, with a species potentially containing multiple subspecies, a subspecies containing multiple strains, and a strain containing multiple (non-identical) genomes. At the other extreme, species with subspecies (’polytypic’) and high diversity are more likely to be free-living generalists with multiple adaptations to distinct and fluctuating environments, with broad geographic ranges or many partially overlapping niches. For example, E. coli has at least six phylogroups that tend to be more prevalent in different habitats. In this context, the general term ‘strain’ could be replaced by the more specific term ecotypes which is an ecologically homogeneous population. A clade within a species that has adapted to a particular environment. The scale of genetic dissimilarity between ecotypes can vary greatly. A mutant within an ecotype can outcompete the other strains in its own ecotype, but not those from a different ecotype. Many researches are aiming to identify the genetic segments (for example, genes, operons, plasmids) that are key to adapting to particular environments and differ within different ecotypes of the same strain.
  4. Although some of the bacteria used in the current study have been approached before for their sensitivity against NPs, we cannot ignore that the bacteria used in the current study may be different ecotypes due to the adaptation to different habitat and different ecological niche.
  5. Previous studies have investigated the use of PVP and PVA as capping aging during the synthesis of CuO nanoparticles [3,4]. However, in the current study, EDTA was implemented as a capping agent, to control the size, shape and to prevent agglomeration of the nanoparticles. In addition, Mg was used as a dopant in CuO nanoparticles due to the comparable ionic radii of Mg2+ ions (0.072 nm) and Cu2+ ions (0.073 nm) [5]. Four different concentrations of Mg dopants were prepared (x=0.005, 0.010, 0.015, and 0.020) in Cu1-xMgxO nanoparticles, taking into account low concentrations to enable their ease incorporation into the lattice, without forming secondary phases.
  6. The current work aimed to investigate the effect of capping undoped and Mg-doped CuO NPs with EDTA on their structural, morphological, and antibacterial properties.

[1] Francis DV, Sood N, Gokhale T. Biogenic CuO and ZnO Nanoparticles as Nanofertilizers for Sustainable Growth of Amaranthus hybridus. Plants 2022, 11, 2776.

[2] Van Rossum T, Ferretti P, Maistrenko OM, Bork P. Diversity within species: interpreting strains in microbiomes. Nature Reviews Microbiology. 2020 Sep;18(9):491-506.

[3] Azharudeen, A.M.; Badhusha, A.M.I.; Khan, M.S.; Prabhu, S.A.; Kumar, P.V.; Karthiga, R.; Odeibat, H.A.; Naz, H.; Buvaneswari, K.; Islam, Md.R. Solar Power Light-Driven Improved Photocatalytic Action of Mg-Doped CuO Nanomaterial Modified with Polyvinylalcohol. J. Nanomater. 2022, 2022, 1–15, doi:10.1155/2022/2430840.

[4] Varghese, R.; Prabu, J.; Johnson, I. Synthesis and Characterization of Mg Doped CuO Nano Particles by Quick Precipitation Method. In; 2017; pp. 159–165 ISBN 978-981-10-2674-4.

[5] Lv, Y.; Li, L.; Yin, P.; Lei, T. Synthesis and Evaluation of the Structural and Antibacterial Properties of Doped Copper Oxide. Dalton Trans. 2020, 49, 4699–4709, doi:10.1039/D0DT00201A.

  • In general, the design of investigation is very similar to one published by Pramothkumar, (ref. 27 of the current manuscript) and some others.

Reply: Based on the authors’ understanding, Pramothkumar et. (2019) have different approach and different biological application as follows:

  1. Although the reference used the same method of preparation, but the doping was totally different from the current study. They used Co, Mn, and Ni with 1.0 wt% only.
  2. They did not use EDTA as capping agent.
  3. They did not investigate any antibacterial activity.
  4. They investigated the degradation of environmental pollutants (4-Nitrophenol and Methylene blue dyes)

Concerning other references, regarding antibacterial activity, as mentioned in the response of comment 1, other authors have used bacteria from different sources. And some are not having different approaches to investigate the antibacterial activity. Generally most of these articles will have the agar well diffusion assay, but others may lack MIC and MBC determination, as well as a time-kill investigation.

  • In addition to the general conclusion about the hopelessness of publication in Molecules, I would like to draw attention to the shortcomings of the study itself in terms of the methods used and the interpretation of the results. The authors do not take into account the possible content of carbon and nitrogen residues from ethylenediamine after the sintering process. Instead, a presence of small amount of carbon in EDX spectra is attributed to carbon tape, used in the measurements. I would suggest to perform TGA analysis before a sintering to elucidate the ethylenediamine decomposition process in course of calcination of nanocomposites.

Reply: The small amount of carbon in the EDX pattern may be attributed to the used carbon tape or residues from EDTA. However, based on TGA measurements from literature, the EDTA decomposes totally at calcination temperatures greater than 400°C [6].

[6] Rajput, A.B.; Hazra, S.; Ghosh, N.N. Synthesis and Characterisation of Pure Single-Phase CoFe2O4 Nanopowder via a Simple Aqueous Solution-Based EDTA-Precursor Route. J. Exp. Nanosci. 2013, 8, 629–639, doi:10.1080/17458080.2011.582170.

  • On TEM images, the size of the NPs looks very confusing. The images attributed by the authors to NPs look more like an organic layer than metal NPs. In addition, the dimensions of NP 41.31; 27.14 etc. cannot be indicated in this way. The method of calculation must be specified, as well as the margin of error. The explanation of the change in the size of nanoparticles by doping with Mg looks very strange and is not confirmed by either other methods or references.

Reply: Regarding the dimensions of the nanoparticles, the distribution of the particle sizes was facilitated by using the ImageJ software, which enabled the size measurement of each particle. Then, the distribution is fitted by a Gaussian function, from which the average size is determined and listed as 41.31 ± 1.76 nm for undoped CuO NPs, 27.14 ± 6.70 nm for x = 0.005 and 33.78 ± 8.54 nm for x = 0.020. The particle size distribution histograms are added to the manuscript as well as the method of determining the average grain sizes. Furthermore, the variations in the average particle size with the doping concentration is discussed and assisted by references (line 113 - line 127 in the manuscript). Moreover, the SEM analysis is modified to account for the size of the nanoparticles, further confirming the size variations with the Mg doping.

  • SEM images (Fig.3) are shown at very different scales that does not allow assessment the similarity or difference between doped and undoped CuO NPs.

Reply: SEM images are replaced by images of the same scale, enabling the comparison between them. Also, a discussion was added to the manuscript (line 131 - line 137).

  • The PL spectra (Fig.5) also look equally. The deconvolutions have been done in an arbitrary manner that have led to unreliable conclusions.

Reply: The deconvolutions of the PL spectra are done using Fityk program, by applying Levenberg-Marquardt fitting method. All the spectra, that seems similar to each other, is compared, taking into account the variation in the intensity and peak positions. The visible part of the spectra is deconvoluted into four Voigt peaks. These peaks are further identified based on the position of the maximum intensity and referred to different visible emissions. The analysis and discussion are adjusted in the manuscript and highlighted in yellow (line 173-183 and line 279-301).

  • In the list of references the titles of journals are missed (see refs. 5,8,9,10,30,43,45, etc). The ref. 11 duplicates the ref. 9. In general, the references list is very sloppy, with some data missed.

Reply: All the references have been checked and modifications were applied.

Note: The line numbers indicated from the manuscript are based on Review-Track Changes-No Markup

Reviewer 2 Report

Comments to the Authors 

This review titled “Synthesis, Characterization, and Antibacterial Activity of Mg- 2

doped CuO Nanoparticles ”, mainly reported the effect of magnesium (Mg) doping on the characteristics and antibacterial properties of copper oxide (CuO) nanoparticles (NPs). Both undoped and doped CuO NPs exhibited significant antibacterial capacities. NPs inhibited the growth of Gram-positive and Gram-negative bacteria. In general, the manuscript is well organized and language is also OK. Therefore, this article can be accepted after minor revision.

 Some small points are shown below.

1. The quality of Fig.5 and 6 need to be improved.

2. The quality of Table should be Three-line Table.

3.  The quality of all figures need to be improved.

Author Response

[Molecules] Manuscript ID: molecules-2072231 – Authors Response

The authors would like to thank the reviewer for his/her precious time, efforts, and invaluable comments. The comments and suggestions were very helpful and had allowed us to explain and improve more aspects of the manuscript. We have carefully addressed all the comments and we are pleased to learn that the manuscript has been substantially improved.

Below is our point-by-point response to the comments as the following:

Reviewer Comments & Authors Responses

This review titled “Synthesis, Characterization, and Antibacterial Activity of Mg-doped CuO Nanoparticles”, mainly reported the effect of magnesium (Mg) doping on the characteristics and antibacterial properties of copper oxide (CuO) nanoparticles (NPs). Both undoped and doped CuO NPs exhibited significant antibacterial capacities. NPs inhibited the growth of Gram-positive and Gram-negative bacteria. In general, the manuscript is well organized and language is also OK. Therefore, this article can be accepted after minor revision.

Some small points are shown below.

  • The quality of Fig.5 and 6 need to be improved.

Reply: Figures 5 and 6 have been modified and improved. The resolution of the figures is now at least 300 Pixels/inch (dpi).

  • The quality of Table should be Three-line Table.

Reply: All tables have been revised and updated to the Three-line format

  • The quality of all figures needs to be improved.

Reply: All figures have been revised and modified. The resolution of the figures is now at least 300 Pixels/inch (dpi).

Reviewer 3 Report

The last two paragraphs of introduction is very confusing. author reports that undoped CuO and Mg doped CuO have been previously studied and effect of EDTA has also been studied. Then what is the novelty of this work. Please explain this in clear words.

2nd page, first paragraph: there are several studies reported on Mg doped CuO References 8-11 as reported. The senstences, "Furthermore, the doped CuO NPs possess better antibacterial activity against Gram-positive bacteria than Gram-negative bacteria, especially against S. aureus. 5% Mg-doped CuO NPs exhibited a bactericidal activity at very low concentrations and their bacteriostatic rate reached 99.9%." NO reference is provided ??? How the present study is different ? 

XRD analysis: There is just an observation of XRD spectra, there is no discussion and comparing with literature, even though, in  introduction, there are several references regarding Mg doping in CuO.  

TEM analysis: No discussion is provided for decrease and further increase in NPs size after doping of Mg with increasing concentration. Author should also correlated these size variation with reference to XRD and TEM analysis.

SEM and EDX analysis: SEM micrographs have not been discussed in the text. EDX analysis shows the presence of Mg (more than 1%) while as per doping it is 0.020....please discuss it clearly 

PL results need to be discussed a little more and Table 1 should be introduced in the text and discussed.

Figure 6 should be broken in two to three part and discussed clearly.

Section 4 should be after the introduction part.

Author should also consider most relevant to the present study and discuss the present results while comparing with other metal oxide nanomaterials such as TiO2 or ZnO:

https://doi.org/10.1016/j.mtsust.2021.100066

https://doi.org/10.1016/j.colsurfb.2017.07.071

Author Response

[Molecules] Manuscript ID: molecules-2072231 – Authors Response

The authors would like to thank the reviewer for his/her precious time, efforts, and invaluable comments. The comments and suggestions were very helpful and had allowed us to explain and improve more aspects of the manuscript. We have carefully addressed all the comments and we are pleased to learn that the manuscript has been substantially improved.

Below is our point-by-point response to the comments as the following:

Reviewer Comments & Authors Responses

  • The last two paragraphs of introduction are very confusing. author reports that undoped CuO and Mg doped CuO have been previously studied and effect of EDTA has also been studied. Then what is the novelty of this work. Please explain this in clear words.

Reply: Although CuO NPs with EDTA have been prepared previously, to the best of our knowledge, the doped CuO NPs reported in the current study have not been prepared with EDTA previously. EDTA as a capping agent may affect the size, morphology, and decreases agglomeration of the doped and undoped CuO NPs. All these features may enhance the antibacterial activity of the NPs as reported in the current study.

The current study is the first to report the effect of undoped and doped CuO NPs on the multidrug-resistant gram-negative S. maltophilia. To the best of our knowledge, and after searching the PubMed database using the following query: ("stenotrophomonas maltophilia"[MeSH Terms] OR ("stenotrophomonas"[All Fields] AND "maltophilia"[All Fields]) OR "stenotrophomonas maltophilia"[All Fields]) AND "maltophilia"[All Fields] AND ("s"[All Fields] AND "maltophilia"[All Fields]) AND "CuO"[All Fields] AND ("nanoparticle s"[All Fields] OR "nanoparticles"[MeSH Terms] OR "nanoparticles"[All Fields] OR "nanoparticle"[All Fields]), Only one very recent study showed up which did not investigate the effect of CuO NPs on S. maltophilia, but investigated the capability of S. maltophilia, isolated from the soil, to synthesis CuO and ZnO nanoparticles [1].

Previous studies have investigated the use of PVP and PVA as capping aging during the synthesis of CuO nanoparticles [3,4]. However, in the current study, EDTA was implemented as a capping agent, to control the size, shape and to prevent agglomeration of the nanoparticles. In addition, Mg was used as a dopant in CuO nanoparticles due to the comparable ionic radii of Mg2+ ions (0.072 nm) and Cu2+ ions (0.073 nm) [5]. Four different concentrations of Mg dopants were prepared (x=0.005, 0.010, 0.015, and 0.020) in Cu1-xMgxO nanoparticles, taking into account low concentrations to enable their ease incorporation into the lattice, without forming secondary phases.

The current work aimed to investigate the effect of capping undoped and Mg-doped CuO NPs with EDTA on their structural, morphological, and antibacterial properties.

[1] Francis DV, Sood N, Gokhale T. Biogenic CuO and ZnO Nanoparticles as Nanofertilizers for Sustainable Growth of Amaranthus hybridus. Plants 2022, 11, 2776.

[3] Azharudeen, A.M.; Badhusha, A.M.I.; Khan, M.S.; Prabhu, S.A.; Kumar, P.V.; Karthiga, R.; Odeibat, H.A.; Naz, H.; Buvaneswari, K.; Islam, Md.R. Solar Power Light-Driven Improved Photocatalytic Action of Mg-Doped CuO Nanomaterial Modified with Polyvinylalcohol. J. Nanomater. 2022, 2022, 1–15, doi:10.1155/2022/2430840.

[4] Varghese, R.; Prabu, J.; Johnson, I. Synthesis and Characterization of Mg Doped CuO Nano Particles by Quick Precipitation Method. In; 2017; pp. 159–165 ISBN 978-981-10-2674-4.

[5] Lv, Y.; Li, L.; Yin, P.; Lei, T. Synthesis and Evaluation of the Structural and Antibacterial Properties of Doped Copper Oxide. Dalton Trans. 2020, 49, 4699–4709, doi:10.1039/D0DT00201A.

  • 2nd page, first paragraph: there are several studies reported on Mg doped CuO References 8-11 as reported. The sentences, "Furthermore, the doped CuO NPs possess better antibacterial activity against Gram-positive bacteria than Gram-negative bacteria, especially against S. aureus. 5% Mg-doped CuO NPs exhibited a bactericidal activity at very low concentrations and their bacteriostatic rate reached 99.9%." NO reference is provided ??? How the present study is different? 

Reply: The reference for the discussed study is added and labelled as reference [7]. The reported study synthesized CuO doped by Mg, Zn, and Ce, via the hydrothermal method. Then, they studied their antibacterial effect against Gram-negative and Gram-positive bacteria.

However, in our study, we have synthesized Mg-doped CuO by the co-precipitation method with EDTA as a capping agent. Then, we have studied the inhibition capacities of the synthesized nanoparticles against various bacteria isolated from the Lebanese wastewater, including Gram-positive bacteria (S. aureus and E. faecium) and Gram-negative bacteria (E. coli and S. maltophilia).

Regarding bacteria, the similarity was only for S. aureus. However, these studies reported a bactericidal effect of Mg-doped CuO NPs but our work revealed a bacteriostatic effect.

The same bacterial species may differ between countries and even between two different areas in the same country. The bacteria used in the current study were isolated from wastewater as mentioned in the Material & Methods section. And to the best of our knowledge, no studies have been carried out to study the effect of NPs on bacteria isolated from wastewater.

According to an elegant recent study by Van Rossum et al. (2020) [2], Many species have been found to have both pathogenic and commensal strains (for example, Escherichia coli and Bacteroides fragilis). Indeed, a classic example are E. coli strains, which can be pathogenic, commensal, host-associated or environmental. The hierarchy of terms used with bacteria, with a species potentially containing multiple subspecies, a subspecies containing multiple strains, and a strain containing multiple (non-identical) genomes. At the other extreme, species with subspecies (’polytypic’) and high diversity are more likely to be free-living generalists with multiple adaptations to distinct and fluctuating environments, with broad geographic ranges or many partially overlapping niches. For example, E. coli has at least six phylogroups that tend to be more prevalent in different habitats. In this context, the general term ‘strain’ could be replaced by the more specific term ecotypes which is an ecologically homogeneous population. A clade within a species that has adapted to a particular environment. The scale of genetic dissimilarity between ecotypes can vary greatly. A mutant within an ecotype can outcompete the other strains in its own ecotype, but not those from a different ecotype. Many researches are aiming to identify the genetic segments (for example, genes, operons, plasmids) that are key to adapting to particular environments and differ within different ecotypes of the same strain.

Although some of the bacteria used in the current study have been approached before for their sensitivity against NPs, we cannot ignore that the bacteria used in the current study may be different ecotypes due to the adaptation to different habitat and different ecological niche.

For the preparation of NPs, these studies showed preparation of NPs by a hydrothermal method. On the other hand, in our study we prepared the NPs by the co-precipitation method. This might have affected the action of the NPs on bacteria, especially the Mg-doped ones.

[2] Van Rossum T, Ferretti P, Maistrenko OM, Bork P. Diversity within species: interpreting strains in microbiomes. Nature Reviews Microbiology. 2020 Sep;18(9):491-506.

  • XRD analysis: There is just an observation of XRD spectra, there is no discussion and comparing with literature, even though, in introduction, there are several references regarding Mg doping in CuO.

Reply: The XRD analysis is further discussed by comparing with literature. The modifications are highlighted in yellow in the manuscript (line 94 - line 103).

  • TEM analysis: No discussion is provided for decrease and further increase in NPs size after doping of Mg with increasing concentration. Author should also correlate this size variation with reference to XRD and TEM analysis.

Reply: The reviewer’s comment has been addressed. And the following paragraph has been added to the discussion (highlighted in the manuscript in yellow (line 119 - line 127)

“These alterations in the average grain sizes with the increase in the concentration of Mg-doping may be attributed to the dissimilarity in Pauling electronegativity that affected the growth rate of Mg-doped CuO nanoparticles. The host Cu ions have a Pauling electronegativity of 1.9, which is higher than that of the doped Mg ions (1.31). This dissimilarity proves the decrease in the growth rate at low concentrations of Mg-doped CuO NPs [15]. However, at higher concentrations, the Mg-doped ions may incorporate into the lattice, not only filling substitutional sites, but also occupying interstitial sites, that yield larger grains, as seen in the sample with x = 0.020.”

[15] Din, S.U.; Sajid, M.; Imran, M.; Iqbal, J.; Shah, B.A.; ullah, M.A.; Shah, S. One Step Facile Synthesis, Characterization and Antimicrobial Properties of Mg-Doped CuO Nanostructures. Mater. Res. Express 2019, 6, 085022, doi:10.1088/2053-1591/ab1c1a.

  • SEM and EDX analysis: SEM micrographs have not been discussed in the text. EDX analysis shows the presence of Mg (more than 1%) while as per doping it is 0.020....please discuss it clearly

Reply: For the SEM analysis: SEM images were replaced by images of the same scale, enabling the comparison between them. Also, a discussion was added to the manuscript (line 131-line 137). For the EDX analysis: In order to compare the theoretical with the experimental percentages in the EDX analysis, we have to check the ratios of Mg (at.%)/Cu (at.%). For this purpose, the discussion of the EDX have been modified and added to the manuscript (line 148-line 150), including this ratio for x = 0.020, Mg/Cu (at.%) = 0.025, which is highly comparable with the theoretical value (x = 0.020).

  • PL results need to be discussed a little more and Table 1 should be introduced in the text and discussed.

Reply: The PL results have been revised, discussed, and added to the manuscript (line 173-line 183 and line 279-line 301). Table 1 have been introduced and discussed in the text. And the following two paragraphs have been added to the manuscript (highlighted in the manuscript in yellow)

“Figure 5(a) shows the room temperature photoluminescence (PL) emission spectra for Cu1-xMgxO NPs with an excitation wavelength of 200 nm. A prominent UV peak appeared at 310 ± 1 nm in all samples, with the highest intensity, as compared to other peaks. Furthermore, the visible emissions in the PL spectra are deconvoluted by four Voigt functions to elucidate the origin of these emissions. The position of the fitted peaks is listed in Table 1, along with the position of the UV peak. It is noticed that the increase in the concentration of the Mg doping in CuO NPs did not affect the position of the peaks, however, it affects their intensity. This is similar to the reported literature, where the doping concentration does not affect the position of the peaks in the visible part of the PL spectra [19–22]. The deconvolution of the PL spectra of Cu1-xMgxO NPs yielded violet (391 ± 1 nm), blue (452 ± 5.5 nm), green (536 nm), and orange-red (628 ± 5 nm) emission peaks.”

“The origin of the UV peak is directly related to the recombination of electron-hole pair, near band gap transition [19,22]. It is noticed that the position of the UV peak is slightly invariant with the doping concentration, however, its intensity increased with Mg-doping. This enhancement of the intensity may be related to the passivation of surface defects that generate radiative recombination [30]. Additionally, the intensity of the UV peak is affected by the electron density and the variation of the morphology and size of the nanoparticles, with the increase of the doping concentration [31]. The visible emissions are highly sensitive to the change in the synthesis conditions, accounting for the type of the dopant and its concentration, the size of the nanoparticle, and its morphology [20]. The size of Mg-doped CuO NPs decreased with the increase of the doping concentration, as noted from TEM and SEM analysis. Hence, the large surface-to-volume ratio stimulates more surface-defect states, as vacancies and interstitials, creating trap levels that radiate visible emissions [20]. Mainly, the intensity of the visible emissions is quenched with the increment of the doping concentration in Cu1-xMgxO NPs, as can be noticed from the inset of Figure 5(a). This decrement in the intensity may be due to the trapping of the photoexcited electron from the conduction band of CuO NPs by the formed deep-level centers from Mg doping [19,31]. The violet and blue emissions are mostly attributed to deep-level defects, indicating the existence of Cu vacancies in the lattice [19]. The green emission was reported to originate from the recombination of single ionized electrons with a photogenerated hole in the valence band, noting the presence of singly ionized oxygen vacancies or dangling bonds of copper [21]. The orange-red emission ascends from the recombination of an electron bound to donor and free holes [20].”

  • Figure 6 should be broken in two to three part and discussed clearly.

Reply: While we understand the reviewer’s concern, and based on our understanding, we would like to highlight that the aim of the figure to see the trend of the results with respect to the four different bacteria used, the undoped and doped CuO NP, and the different at.%. Breaking this figure may not clearly illustrate the trend of the results among the investigated parameters. The discussion of this results is highlighted in yellow (line 352 - line 361 in the manuscript).

  • Section 4 should be after the introduction part.

Reply: The authors are happy to respond to the reviewer comment, but the journal guidelines state that the “Materials and Methods” should be after the “Discussion” and before the “Conclusion”.

  • Author should also consider most relevant to the present study and discuss the present results while comparing with other metal oxide nanomaterials such as TiO2 or ZnO: https://doi.org/10.1016/j.mtsust.2021.100066, https://doi.org/10.1016/j.colsurfb.2017.07.071

Reply: The comment has been addressed and the changes have been done in the discussion (line 338-line 341 in the manuscript).

Note: The line numbers indicated from the manuscript are based on Review-Track Changes-No Markup

Round 2

Reviewer 1 Report

I have no more comments

Reviewer 3 Report

Authors have responded very well and modifications have been done. It can be accepted now.